# Tracing Back the Malicious Clients in Poisoning Attacks to Federated Learning

**Yuqi Jia**
Duke University
yuqi.jia@duke.edu

**Minghong Fang**
University of Louisville
minghong.fang@louisville.edu

**Hongbin Liu**
Duke University
hongbin.liu@duke.edu

**Jinghuai Zhang**
University of California, Los Angeles
jinghuai1998@g.ucla.edu

**Neil Gong**
Duke University
neil.gong@duke.edu

## Abstract

Poisoning attacks compromise the training phase of federated learning (FL) such that the learned global model misclassifies attacker-chosen inputs called *target inputs*. Existing defenses mainly focus on protecting the training phase of FL such that the learnt global model is poison free. However, these defenses often achieve limited effectiveness when the clients' local training data is highly non-iid or the number of malicious clients is large, as confirmed in our experiments. In this work, we propose *FLForensics*, the first *poison-forensics* method for FL. FLForensics complements existing training-phase defenses. In particular, when training-phase defenses fail and a poisoned global model is deployed, FLForensics aims to trace back the malicious clients that performed the poisoning attack after a misclassified target input is identified. We theoretically show that FLForensics can accurately distinguish between benign and malicious clients under a formal definition of poisoning attack. Moreover, we empirically show the effectiveness of FLForensics at tracing back both existing and adaptive poisoning attacks on five benchmark datasets. Our code and data are available at: `https://github.com/jyqhahah/FLForensics`.

## 1 Introduction

Federated learning (FL) [32] is a distributed learning paradigm, allowing many clients jointly train a *global model* without sharing raw data. Specifically, in each round the server broadcasts the current model, clients update it on their private data, and the server aggregates the updates [32]. FL has been widely deployed in various real-world applications, such as credit risk prediction [2] and next-word prediction [1]. However, FL's distributed updates make it prone to poisoning attacks: *malicious clients* can submit crafted updates that the server accepts [16, 4, 37, 40, 8, 19]. The resulting global model maps an attacker-chosen *target input* to an attacker-chosen *target label* while leaving other predictions intact. This target input may be any sample carrying an injected trigger (a backdoor) or even a clean sample without a trigger.

Existing defenses [42, 7, 33, 17, 43, 20, 21] against poisoning attacks to FL focus on *protecting the training phase*. Robust aggregators such as Trim, Median [42], FLTrust [7], and FLAME [33] try to filter potentially malicious updates, while detectors like FLDetector detects clients whose updates are inconsistent across multiple rounds [43]. However, these training-phase defenses are insufficient. In particular, when data are highly non-IID or attackers control many clients, malicious and benign updates become hard to distinguish, as shown in our experiments. Consequently, even if these training-phase defenses are adopted, the learnt global model may still be poisoned and the poisoned global model is deployed.

39th Conference on Neural Information Processing Systems (NeurIPS 2025).

**Our work:** In this work, we propose *FLForensics*, the first *poison-forensics* method for FL. Unlike training-phase defenses, FLForensics aims to trace back the malicious clients that performed the poisoning attack after the attack has happened, i.e., after training-phase defenses fail, a poisoned global model has been deployed, and a misclassified target input has been identified. Identifying such a misclassified input—e.g., via manual inspection or automatic tools [22, 14, 31]—is orthogonal to FLForensics. For instance, we show that our FLForensics can be adapted to detect whether a misclassified input is a misclassified target input or not in Appendix G. FLForensics consists of two major steps: *calculating influence scores* and *detecting malicious clients*. Step I assigns each client an influence score for the misclassified *target input*, and Step II uses these scores to distinguish *malicious* from benign clients.

**Calculating influence scores.** To quantify the misclassification of a target input, we measure each client's effect by the change it causes in the global model's cross-entropy loss on the misclassified *target input* across all rounds. One challenge is that clients' local training data are often non-iid. Specifically, some benign clients (denoted as *Category I*) have a large amount of local training examples with the target label, while other benign clients (*Category II*) do not. As a result, both malicious clients and Category I benign clients have large influence scores, making it challenging to distinguish them. To separate them, the server also tests every client on a random *non-target input* with the target label, yielding another influence score. Thus client $i$ receives a two-dimensional score $(s_i, s_i')$ from the target and non-target inputs, respectively.

**Detecting malicious clients.** We observe that malicious clients have large $s_i$ but small $s_i'$, Category I benign clients have large $(s_i, s_i')$, and Category II benign clients have small $(s_i, s_i')$. Based on such observations, we detect malicious clients by clustering their 2-D scores with HDBSCAN [6], which needs no preset cluster count. We further use a *scaled Euclidean distance*, which normalizes the two dimensions of a two-dimensional influence score to have the same, comparable scale. Clusters with positive mean $s_i$ become *potentially malicious*. However, these clusters may also include Category I benign clients. To address the challenge, our key observation is that the influence-score gap $s_i' - s_i$ of a malicious client is smaller than that of a Category I benign client. FLForensics leverages this to pinpoint the truly *malicious* clients inside each cluster.

**Theoretical and empirical evaluation.** Theoretically, we show the security of FLForensics against poisoning attacks. In particular, based on a formal definition of poisoning attacks and mild assumptions, we show that 1) both malicious clients and Category I benign clients have larger influence scores $s_i$ than Category II benign clients, and 2) a malicious client has a smaller influence-score gap $s_i' - s_i$ than a benign client. Empirically, we comprehensively evaluate FLForensics on five benchmark datasets. Our results show that FLForensics can accurately trace back malicious clients under various existing and adaptive attacks. We note that training-phase defenses are ineffective for most attack scenarios in our evaluation.

We summarize our main contributions as follows:

- We propose the first poison-forensics method called FLForensics to trace back malicious clients in FL.
- We theoretically show the security of FLForensics against poisoning attacks.
- We empirically evaluate FLForensics on five benchmark datasets against existing and adaptive poisoning attacks.

## 2 Preliminaries and Related Work

**Federated learning (FL):** FL enables $n$ clients to collaboratively train a shared *global model* under a central server's coordination. Each client updates the global model using its local data and sends a model update to the server, which aggregates them (e.g., via FedAvg [32]) to update the global model:

$$w_{t+1} = w_t + \alpha_t \cdot Agg(g_t^{(1)}, g_t^{(2)}, \cdots, g_t^{(n)}), \tag{1}$$

where $\alpha_t$ is the learning rate. In practice, only a subset of clients is selected per round. Many FL variants [32, 42, 7, 33, 28, 11, 39, 43, 18] differ primarily in their aggregation rules.

**Poisoning attacks to FL:** Poisoning attacks aim to corrupt the training process so that the resulting global model is compromised. In *targeted poisoning attacks* [3, 37], the model misclassifies attacker-

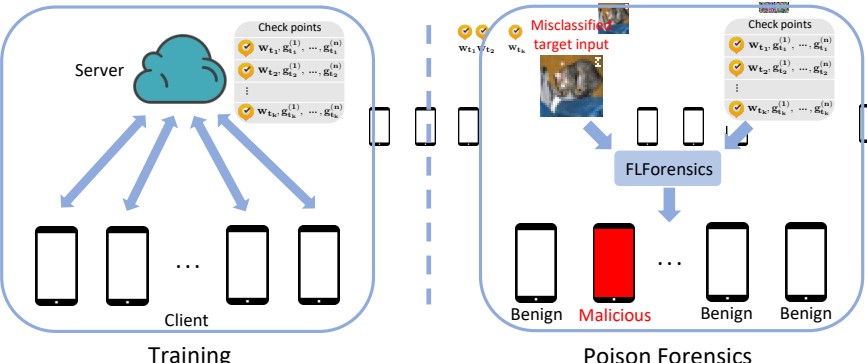

Figure 1: Overview of FLForensics. During training, the server stores the intermediate global models and clients' model updates in some training rounds called *check points*. Given a misclassified target input detected after deploying the poisoned global model, the server uses FLForensics to trace back the malicious clients that performed the poisoning attack.

chosen inputs into a target label while maintaining overall accuracy. We refer to these simply as poisoning attacks. Some attacks use *trigger-embedded* target inputs (i.e., backdoor attacks [4, 3]), where any input with a trigger is misclassified. Others use *triggerless* target inputs [37], which are naturally occurring but mislabeled edge cases. In both cases, malicious clients manipulate their local data or model updates to implant the attack during training. Details are deferred to Appendix A.1.

**Training-phase defenses:** Most defenses aim to secure the training phase to prevent a poisoned global model. Some approaches improve the aggregation rule to tolerate malicious updates, e.g., Trimmed Mean, Median [42], FLTrust [7], and FLAME [33]. Others [9, 11] offer provable guarantees by training multiple global models and using ensemble prediction. See Appendix A.2 for more details. Furthermore, some defenses focus on detection and recovery from attacks. FLDetector [43] detects clients with inconsistent model updates, and FedRecover [10] reconstructs a clean model without retraining from scratch.

However, these training-phase defenses suffer from a few key limitations. Robust FL methods still struggle when malicious clients are numerous or client data is highly non-iid. Moreover, FLDetector cannot detect data poisoning attacks where malicious clients poison data but follow protocol. Consequently, a poisoned model may still be deployed. In our work, we assume a poisoned global model is already deployed. Given a misclassified target input detected post-deployment, our goal is to trace back the malicious clients responsible for the attack.

**Poison forensics for centralized learning:** Poison forensics methods [36, 25, 13] trace the source of misclassification in centralized learning. PF [36] and GAS [25] identify poisoned training data responsible for a misclassification, while Beagle [13] recovers triggers from multiple poisoned inputs. These methods assume centralized access to training data and do not generalize well to FL, as shown in our experiments. In contrast, our FLForensics is the first poison-forensics method tailored to FL, capable of identifying malicious clients post-deployment. Interestingly, our method also performs well in centralized settings with unbalanced data, where existing methods degrade (see Appendix G).

## 3 Threat Model

**Poisoning attacks:** We assume an attacker compromises the FL training by controlling a set of *malicious clients*, which may be fake or compromised genuine ones. These clients craft model updates that poison the global model, while the server remains honest. After training, the poisoned model is deployed for real-world use. In this work, we focus on *targeted* poisoning attacks, where the model misclassifies attacker-chosen *target inputs* as an attacker-specified *target label*, while behaving normally on other inputs. Appendix I provides discussion of untargeted poisoning attacks.

**Poison forensics:** We adopt a standard poison-forensics setting [36, 25, 13]: a misclassified target input in a poisoning attack is detected after model deployment. Detection can be done automatically [22, 14, 31] or manually by users observing application errors caused by the misclassification.

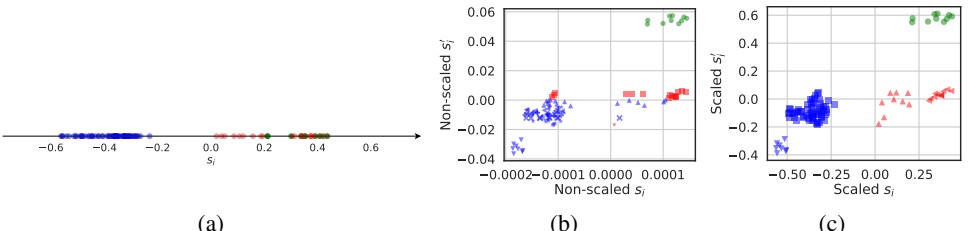

|     | (a) | (b) | (c) |

Figure 2: (a) Influence scores $s_i$ and (b–c) clustering results in one of our experiments using Euclidean and scaled Euclidean distance. Dots represent clients: red (malicious), green (Category I benign), and blue (Category II benign). Different *markers* represent different HDBSCAN clusters.

Given such a misclassified target input, the goal is to trace back the malicious clients responsible for the attack. In Section 7, we further show that FLForensics can help determine whether a misclassified input is indeed a target input.

## 4  Our FLForensics

### 4.1  Overview

During training, FLForensics stores intermediate global models and client updates as *check points*. When a target input is misclassified, FLForensics proceeds in two steps: (1) it computes each client's *influence score* from the stored check points, and (2) it detects malicious clients via clustering the influence scores. Figure 1 shows the workflow and Algorithm 1 provides the pseudo-code.

### 4.2  Calculating Influence Scores

**Quantifying a misclassification:**  We denote a target input as $x$, which is misclassified as the target label $y$ by the poisoned global model $w$. To measure a client's contribution to misclassification, we use the cross-entropy loss $\ell_{CE}(x, y; w)$ of the poisoned global model $w$. Denote $w_t$ as the global model in training round $t$, where $t = 1, 2, \cdots, R$, then $w_0$ is the initial global model and $w_R$ is the final model after $R$ rounds. The overall loss change is $\ell_{CE}(x, y; w_0) - \ell_{CE}(x, y; w_R)$, which we attribute to clients' model updates across rounds.

**Expanding the training process:**  Since the loss change is accumulated over the $R$ training rounds, we expand them to measure the influence of each client on the loss change. Specifically, according to Taylor expansion, for training round $t$, we have $\ell_{CE}(x, y; w_{t+1}) \approx \ell_{CE}(x, y; w_t) + \nabla \ell_{CE}(x, y; w_t)^\top (w_{t+1} - w_t)$. Therefore, by summing over $R$ training rounds, we have:

$$\ell_{CE}(x, y; w_0) - \ell_{CE}(x, y; w_R) \approx -\sum_{t=1}^{R} \nabla \ell_{CE}(x, y; w_t)^\top (w_{t+1} - w_t). \tag{2}$$

**Assigning influence scores:**  If a client is selected in training round $t$, we quantify its contribution to the model difference $w_{t+1} - w_t$. Then we obtain an influence score for it by summing over such contributions over multiple training rounds that involve this client. Let $C_t$ be the set of selected clients in round $t$. If we we assume the global model is updated as if using only the model update $g_t^{(i)}$ of client $i$, then $w_{t+1} - w_t \approx \alpha_t \cdot g_t^{(i)}$, and $i$'s influence score is:

$$s_i = -\sum_{t \text{ s.t. } i \in C_t} \alpha_t \nabla \ell_{CE}(x, y; w_t)^\top g_t^{(i)}. \tag{3}$$

**Using check points to save space and computation:**  If FLForensics uses all training rounds to calculate the influence scores, the server needs to save global models and clients' updates in all training rounds, which incurs substantial space and computation overhead. To reduce overhead, we compute influence scores using a subset of $k$ *check points* $\Omega = \{t_1, ..., t_k\}$:

$$s_i = -\sum_{t \in \Omega \text{ s.t. } i \in C_t} \alpha_t \nabla \ell_{CE}(x, y; w_t)^\top g_t^{(i)}. \tag{4}$$

---

**Algorithm 1** FLForensics

---

**Input:** Misclassified target input $x$, target label $y$, non-target input $x'$, check points $\Omega = \{t_1, t_2, \cdots, t_k\}$, global models $\{w_t\}_{t \in \Omega}$ in the check points, selected clients $C_t$ in each check point $t \in \Omega$, and clients' model updates $\{g_t^{(i)}\}_{t \in \Omega, i \in C_t}$.

**Output:** Predicted malicious clients $\mathcal{M}$.

1: //Calculating influence scores
2: **for** $i = 1$ to $n$ **do**
3:      $s_i = -\sum_{t \in \Omega \text{ s.t. } i \in C_t} \alpha_t \nabla \ell_{CE}(x, y; w_t)^\top g_t^{(i)}$;
4:      $s_i' = -\sum_{t \in \Omega \text{ s.t. } i \in C_t} \alpha_t \nabla \ell_{CE}(x', y; w_t)^\top g_t^{(i)}$;
5: **end for**
6: $\mathcal{I} \leftarrow \{(s_1, s_1'), (s_2, s_2'), \cdots, (s_n, s_n')\}$;
7: //Detecting malicious clients
8: $c_1, \cdots, c_m, c_{outlier} \leftarrow \text{HDBSCAN}(\mathcal{I})$;        $\triangleright$ $c_{outlier}$ is a set that contains all outliers if any.
9: $c_{p_1}, c_{p_2}, \cdots, c_{p_v} \leftarrow$ clusters with average $s_i > 0$;        $\triangleright$ Get potential malicious clusters.
10: $threshold \leftarrow \sum_{j=1}^{v} \sum_{i \in c_{p_j}} s_i' / \sum_{j=1}^{v} \sum_{i \in c_{p_j}} s_i$;
11: $\mathcal{M} = \emptyset$;
12: **for** $j = 1$ to $v$ **do**
13:      $threshold_j \leftarrow \sum_{i \in c_{p_j}} s_i' / \sum_{i \in c_{p_j}} s_i$;
14:      **if** $threshold_j \leq threshold$ **then** $\mathcal{M} = \mathcal{M} \cup c_{p_j}$;
15:      **end if**
16: **end for**
17: **for** $i$ in $c_{outlier}$ **do**
18:      $threshold_i \leftarrow s_i'/s_i$;
19:      **if** $threshold_i \leq threshold$ **then** $\mathcal{M} = \mathcal{M} \cup \{i\}$;
20:      **end if**
21: **end for**
22: **return** $\mathcal{M}$;

---

The space cost is linear to the model size, number of check points, and active clients per round.

**Using a non-target input to augment clients' influence scores:** Due to non-iid data, some benign clients (denoted as *Category I*) may have many examples labeled $y$ and thus yield high $s_i$, similar to malicious clients. Other benign clients (*Category II*) do not, and yield lower scores. In particular, our theoretical analysis in Appendix B shows that malicious clients and Category I benign clients both have larger influence scores than Category II benign clients. Figure 2a shows the influence scores $s_i$ of malicious, Category I benign, and Category II benign clients in an experiment. To differentiate them, we compute a second influence score using a *non-target input $x'$*:

$$s_i' = - \sum_{t \in \Omega \text{ s.t. } i \in C_t} \alpha_t \nabla \ell_{CE}(x', y; w_t)^\top g_t^{(i)}. \tag{5}$$

$x'$ can be either a random or true input, and we show in Appendix G that both choices yield similar performance. This gives each client a two-dimensional influence score $(s_i, s_i')$. Typically, Category I benign clients have both scores large, while malicious clients have a large $s_i$ but a small $s_i'$.

### 4.3 Detecting Malicious Clients

Our method is based on two observations: (**I**) Both malicious and Category I benign clients have larger $s_i$ than Category II clients. (**II**) Malicious clients have smaller gaps $s_i' - s_i$ than benign clients. We provide theoretical justification for these observations in Appendix B.

**Clustering the clients via HDBSCAN with scaled Euclidean distance:** Let $\mathcal{I} = \{(s_1, s_1'), ..., (s_n, s_n')\}$ denote client scores. We cluster clients using HDBSCAN [6], which does not require specifying the number of clusters and handles outliers. To account for differences in score scales, we normalize each score dimension by its range and compute scaled Euclidean distance, i.e., $s_i$ as $s_i / (\max_{j=1}^{n} s_j - \min_{j=1}^{n} s_j)$, and $s_i'$ as $s_i' / (\max_{j=1}^{n} s_j' - \min_{j=1}^{n} s_j')$. Figures 2b and 2c show the improved separation using this distance metric.

**Identifying malicious clients and Category I benign clients based on Observation I:** We treat clusters with positive average $s_i$ as *potentially malicious*, since both malicious and Category I benign clients can fall into this category. For instance, in Figure 2c, both red clusters (malicious) and the green cluster (Category I benign) are potential malicious clusters.

**Distinguishing between malicious clients and Category I benign clients based on Observation II:** To further separate malicious and Category I benign clients, we compute the ratio of average $s_i'$ to $s_i$ in each potential malicious cluster. Based on Observation II, a cluster is classified as malicious if this ratio is below a *threshold*. We set it to the mean ratio across all such clusters. Furthermore, HDBSCAN may output some clients as outliers that do not belong to any cluster. Outlier clients are handled similarly by comparing their $s_i'/s_i$ to the same threshold. Figure 2c illustrates how this approach identifies the red clusters as malicious.

## 5    Experiments

### 5.1    Experimental Setup

**Datasets:** We conduct our experiments using five diverse benchmark datasets: four image datasets (CIFAR-10, Fashion-MNIST, MNIST, and ImageNet-Fruits) and one text dataset (Sentiment140). Detailed descriptions of these datasets are provided in Section C in Appendix.

**FL training settings:** Following [7, 16], we model FL with 100 clients. For CIFAR-10, Fashion-MNIST, MNIST, and ImageNet-Fruits we create non-IID partitions using the method of [16] (Appendix F shows the details). Since Sentiment140 already exhibits user-level non-IID, we simply group users uniformly at random into 100 clients. We train a ResNet-20 [26] for CIFAR-10, a CNN (Table 4 in Appendix) for Fashion-MNIST and MNIST, a LSTM [27] for Sentiment140, and a ResNet-50 [26] for ImageNet-Fruits. Default hyper-parameters (learning rate, batch size, global rounds, local epochs) are listed in Table 5 in Appendix. By default, every round involves all clients and the server aggregates via FedAvg [32]. We will also conduct experiments to vary the client fraction and the aggregation rule to test FLForensics.

**Poisoning attacks to FL:** We consider three popular poisoning attacks to FL, i.e., *Scaling* [3], *A little is enough (ALIE)* [4], and *Edge* [37] attacks. Scaling and ALIE use trigger-embedded target inputs, while Edge uses triggerless target inputs. The description of those attacks are shown in Appendix D. By default, we assume there are 20% malicious clients, who perform attacks in each training round. We will also explore the impact of the fraction of malicious clients and fraction of attacked training rounds on FLForensics.

**Evaluation metrics:** We evaluate with *detection accuracy (DACC)*, *false positive rate (FPR)*, and *false negative rate (FNR)*. DACC is the fraction of clients correctly classified; FPR the fraction of benign clients classified as malicious; FNR the fraction of malicious clients missed. Higher DACC and lower FPR/FNR indicate a better method. We report the *attack success rate (ASR)* in Table 9, which means the fraction of target inputs that the poisoned model predicts as the target label.

**Poison-forensics settings:** For each dataset we randomly choose a misclassified *target input*. FLForensics also requires a non-target input. Because the server may lack real data, it synthesizes a *non-target input*: image pixels are sampled i.i.d. from $U(0, 1)$, and text is a random tweet of the same length. Results in Appendix G also show that using a true input instead can slightly improves FLForensics. By default, the server saves the global model and clients' updates every 10 rounds. Each update $g_t^{(i)}$ is $\ell_2$-normalized before computing influence scores in Equations 4 and 5 to offset scale differences. Unless stated otherwise, we report results on CIFAR-10 under Scaling attack. All the experiments are finished on one single Quadro RTX 6000 GPU with 24GB memory.

### 5.2    Compared methods

We compare FLForensics with the following methods including variants of FLForensics.

**Poison Forensics (PF) [36].** PF is designed for centralized learning. To apply it in FL, we assume the server can access clients' local data. PF identifies poisoned training examples, and a client is classified as malicious if its fraction of detected poisoned samples exceeds the average across clients.

Table 1: DACC/FPR/FNR of FLDetector, a training-phase method to detect malicious clients.

| Attack | MNIST | Fashion-MNIST | CIFAR-10 | Sentiment140 | ImageNet-Fruits |
|---|---|---|---|---|---|
| Scaling | 0.960/0.038/0.050 | 0.870/0.138/0.100 | 0.400/0.538/0.850 | 0.020/0.975/1.000 | 0.475/0.438/0.875 |
| ALIE | 0.000/1.000/1.000 | 0.000/1.000/1.000 | 0.000/1.000/1.000 | 0.010/0.988/1.000 | 0.075/0.906/1.000 |
| Edge | 0.160/0.800/1.000 | 0.390/0.563/0.800 | 0.160/0.800/1.000 | 0.080/0.900/1.000 | 0.750/0.281/0.250 |

Table 2: Results of FLForensics and compared poison-forensics methods.

| Attack | Method | Dataset (DACC/FPR/FNR) | | | | |
|---|---|---|---|---|---|---|
| | | MNIST | Fashion-MNIST | CIFAR-10 | Sentiment140 | ImageNet-Fruits |
| Scaling | PF | 0.900/0.125/0.000 | 0.900/0.125/0.000 | 0.900/0.125/0.000 | **0.990/0.013/0.000** | 0.600/0.375/0.500 |
| | FLForensics-G | 0.480/0.413/0.950 | 0.740/0.100/0.900 | 0.900/0.125/0.000 | **0.990/0.013/0.000** | 0.600/0.313/0.750 |
| | FLForensics-A | 0.900/0.125/0.000 | **1.000/0.000/0.000** | 0.900/0.125/0.000 | 0.900/0.125/0.000 | 0.900/0.125/0.000 |
| | FLForensics | **1.000/0.000/0.000** | **1.000/0.000/0.000** | **1.000/0.000/0.000** | 0.980/0.025/0.000 | **1.000/0.000/0.000** |
| ALIE | PF | 0.740/0.325/0.000 | 0.900/0.125/0.000 | 0.900/0.125/0.000 | 0.760/0.275/0.100 | **1.000/0.000/0.000** |
| | FLForensics-G | 0.520/0.363/0.950 | 0.740/0.100/0.900 | 0.900/0.125/0.000 | 0.980/0.025/0.000 | 0.525/0.406/0.750 |
| | FLForensics-A | 0.900/0.125/0.000 | **1.000/0.000/0.000** | 0.900/0.125/0.000 | 0.900/0.125/0.000 | 0.875/0.156/0.000 |
| | FLForensics | **1.000/0.000/0.000** | **1.000/0.000/0.000** | **1.000/0.000/0.000** | **1.000/0.000/0.000** | **1.000/0.000/0.000** |
| Edge | PF | 0.920/0.100/0.000 | **1.000/0.000/0.000** | 0.820/0.225/0.000 | **0.970/0.038/0.000** | 0.850/0.188/0.000 |
| | FLForensics-G | 0.930/0.088/0.000 | 0.920/0.100/0.000 | 0.920/0.100/0.000 | 0.940/0.075/0.000 | 0.700/0.125/1.000 |
| | FLForensics-A | 0.920/0.100/0.000 | 0.920/0.100/0.000 | 0.910/0.113/0.000 | 0.920/0.100/0.000 | 0.800/0.125/0.500 |
| | FLForensics | **1.000/0.000/0.000** | **1.000/0.000/0.000** | **0.980/0.000/0.100** | **0.970/0.038/0.000** | **0.950/0.031/0.125** |

**FLForensics-G (GAS [25] + FLForensics).** GAS computes influence scores for training examples in centralized learning. However, it lacks a detection step. We extend it to FL (with access to local data) and combine it with FLForensics as an end-to-end method. Specifically, we compute influence scores using GAS and then detect poisoned examples using HDBSCAN (details in Appendix E). Similar to PF, clients are marked as malicious based on their fraction of detected poisoned examples.

**FLForensics-A.** This is a variant of FLForensics that uses only a target input $x$. The server computes influence scores $s_i$ for each client (Equation 4), clusters clients with HDBSCAN, and treats clusters with positive average scores as malicious. We include this variant to highlight the limitations of using only a target input.

### 5.3 Experimental Results

**Training-phase defenses are insufficient:** Training-phase defenses rely on robust aggregation or attacker detection to prevent poisoning attacks. However, Table 9 in Appendix shows robust FL methods, such as Trim, Median [42], FLTrust [7], and FLAME [33], still leave high attack-success rates. Table 1 further shows that FLDetector often misses malicious clients. Non-iid data blur the line between benign and malicious updates, so these defenses remain vulnerable.

**FLForensics is effective and outperforms baselines:** Table 2 shows the results of FLForensics and compared poison-forensics methods. FLForensics accurately traces attackers across all datasets and attacks: its DACC is always 1 or close to 1, while both FPR and FNR remain at 0 or below 3% in only a few Scaling and Edge attack cases. Furthermore, FLForensics outperforms all compared forensics methods. FLForensics-A, which relies solely on the target input, mislabels >10% of benign clients on CIFAR-10. **This shows that using only target input $x$ cannot effectively distinguish between malicious and Category I benign clients.** PF and FLForensics-G, even when given the unfair advantage of direct access to local data, still trail behind FLForensics. When using clients' updates, as shown in Table 10 in Appendix, FLForensics-G performs worse, while PF achieves nearly the same accuracy it attains with direct access to local data.

**Other experiments:** We conduct additional studies to evaluate the robustness and versatility of FLForensics. Details are shown in Appendix G. First, we compare using a random input versus a true target-class input as the non-target input in FLForensics; results are generally comparable, with slightly lower FPRs when using a true input. Second, we apply FLForensics to clean-label attacks,

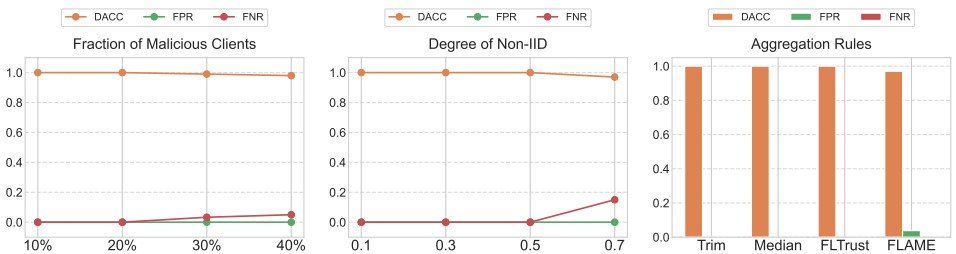

(a) Fraction of malicious clients      (b) Degree of non-iid      (c) Aggregation rules

Figure 3: Ablation-study results for FLForensics. Figure 7 in the Appendix shows additional studies (e.g., check points, client fraction, and scaling factor).

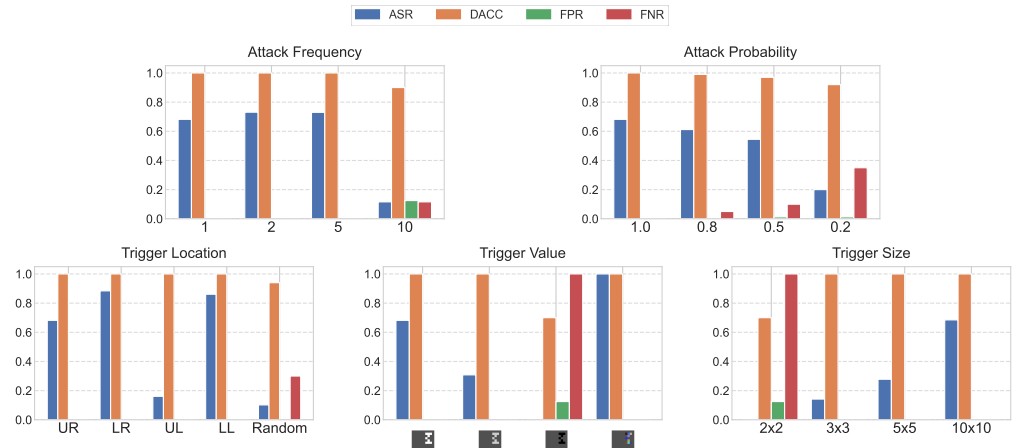

Figure 4: Results of FLForensics for adaptive attacks.

where poisoned samples retain their original labels. Even in this setting, FLForensics performs well (e.g., DACC=0.95, FPR=0.06, FNR=0.0 on CIFAR-10). Lastly, we show that FLForensics can be adapted to centralized learning by treating each training example as a pseudo-client. On imbalanced datasets, it outperforms existing poison-forensics baselines, similar to the FL case with non-iid data.

## 5.4 Ablation Studies

This section presents several ablation studies for FLForensics, including the impact of (i) the fraction of malicious clients, (ii) degree of non-IID data, and (iii) aggregation rules. Appendix H further shows the impact of the number of check points, scaling factor, and fraction of selected clients.

**Impact of fraction of malicious clients:** Figure 3a shows that FLForensics works well even when a large fraction of clients (e.g., 40%) are malicious. Specifically, FLForensics achieves the FNR≤5% and FPR=0 for attacker fractions varies from 10% to 40%. We consider at least 10% of malicious clients because the attacks themselves become ineffective [16] when the fraction is small.

**Impact of degree of non-iid:** Based on Figure 3b, FLForensics works well across different degrees of non-iid. In particular, FLForensics achieves 0 FPR and at most 15% FNR, confirming its design for non-IID data. Note that 0.1 degree of non-iid represents the iid setting.

**Impact of aggregation rule:** Figure 3c shows that FLForensics still identifies malicious clients when the server uses Byzantine-robust aggregation rules. Although these rules alone cannot stop the attacks (as shown in Table 9), pairing them with FLForensics provides strong defense. FLForensics achieves slightly higher FPR under FLAME because Scaling attack is less effective for FLAME, causing a few Category I clients to be misclassified as malicious.

# 6 Adaptive Attacks

Our theoretical analysis in Appendix B proves FLForensics resilient to any (adaptive) poisoning attack that meets our formal definition. Attacks that violate this definition can hurt FLForensics's detection but usually decrease their own ASR. We thus test several such adaptive variants under default settings, e.g., CIFAR-10, 150 check points, and all clients participate in each round.

**Attack frequency and probability:** To evade detection, malicious clients may attack only intermittently. First, they can strike every $e$ training rounds. As shown in the subfigures on the first row of Figure 4, when $e \leq 5$, attacks remain effective (high ASR) and FLForensics is still effective; once $e \geq 10$, ASR falls below 12% and FLForensics's performance drops because the threat itself is weak. Second, clients may attack each round with probability $p$. As shown in Figure 4, when $p$ declines, ASR and FLForensics's recall both decrease, but FLForensics stays effective for $p \geq 0.5$ with a FPR less than 1.25%. Even at $p = 0.2$, while the ASR is 20%, FLForensics misses some malicious clients but misclassifies at most one benign client. This means if an attacker aims to evade FLForensics, its attack becomes less or not effective.

**Trigger location, value, and size:** Subfigures on the second row of Figure 4 evaluate FLForensics under the Scaling attack as we vary the trigger's *location*, *value*, and *size*. Attacks with triggers placed at fixed locations—UR, LR, UL, and LL—achieve high ASRs, where these denote the upper right, lower right, upper left, and lower left corners of the image, respectively. The trigger in these experiments is the same as in [3]. In these cases, FLForensics identifies all malicious clients. When the trigger is at random locations, the attack is less effective, leading to some missed detections (FNR around 30%). Furthermore, altering the RGB value shows a similar pattern: once the attack is effective—e.g., white $(255, 255, 255)$ or gray $(204, 204, 204)$ squares—FLForensics traces every malicious client, whereas an all-black trigger, which fails to poison the model (ASR=0), leaves them undetected. Finally, we find that increasing trigger size boosts ASR: a $2 \times 2$ square is ineffective and hampers detection, but sizes larger than $3 \times 3$ already let FLForensics catch every malicious client even when ASR is only 14.2%. Overall, FLForensics succeeds whenever the trigger is large, distinctly colored, or consistently placed enough to mount a meaningful attack, and it degrades only when the attack itself has little impact.

# 7 Discussion

**Recovering from attacks after FLForensics:** After FLForensics detects malicious clients, the server can discard their updates and re-train the global model. On CIFAR-10 under Edge attack, the case with highest FNR, test accuracy improves from 81.9% to 82.8%, while ASR drops from 17.4% to 5.6%. This recovery can be made communication-efficient with methods like FedRecover [10].

**Detecting misclassified target input:** FLForensics assumes the given misclassified sample is a target input. If it is not, the real attackers may not contribute to it and thus escape detection. We adapt FLForensics to first decide whether a misclassified sample is a target input. For a misclassified input $x$ and a non-target input $x'$, we compute each client's influence scores $(s_i, s_i')$ and run FLForensics to form *potential malicious* clusters $c_{p_j}$. Our intuition is that if all potential malicious clusters have nearly identical mean $s_i$ and $s_i'$, making $x$ and $x'$ indistinguishable in influence, we label the misclassified input $x$ as non-target. If every such cluster satisfies $c_{p_j}$ satisfies $\alpha \leq \sum_{i \in c_{p_j}} s_i' / \sum_{i \in c_{p_j}} s_i \leq \frac{1}{\alpha}$ for some $\alpha < 1$, we judge $x$ to be a non-target input; otherwise, we treat it as a target input. With $\alpha = 0.2$, we evaluate the method on 50 randomly chosen target samples and 50 misclassified non-target samples. It correctly labeled 96% of the target inputs and 98% of the non-target inputs.

# 8 Conclusion and Future Work

In this work, we propose FLForensics, the first poison-forensics method to trace back malicious clients in FL. We theoretically show the security of FLForensics against (adaptive) poisoning attacks under a formal definition of poisoning attack. Moreover, our empirical evaluation results on multiple benchmark datasets show that FLForensics can accurately trace back malicious clients against both state-of-the-art and adaptive poisoning attacks. An interesting future work is to extend FLForensics

to untargeted poisoning attacks and explore the security of FLForensics against strategically crafted misclassified target input.

## Acknowledgement

We thank the anonymous reviewers for their constructive comments. This work was supported by NSF under grant no. 2131859, 2125977, 2112562, and 1937787.

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

# Appendix

## A    Related Work

### A.1    Details of Poisoning Attacks in FL

**Trigger-embedded poisoning attacks (backdoor attacks):**  These attacks treat any input embedded with a specific trigger as a target input. In the **Scaling attack** [3], malicious clients duplicate local data, embed a trigger into these examples, relabel them to the target label, and amplify the resulting model update with a scaling factor $\lambda$ before sending it to the server.

The **ALIE attack** [4] follows a similar data manipulation strategy, but constructs adversarial updates by solving an optimization problem to maximize the malicious effect.

**Triggerless poisoning attacks:**  These attacks target specific inputs without any trigger. For example, in the **Edge-case attack** [37], the attacker injects out-of-distribution samples (edge cases) into the local training set of malicious clients and labels them with the target label. These inputs become target inputs post-training due to label manipulation.

### A.2    Details of Training-phase Defenses

**Robust aggregation:**  Byzantine-robust methods like **Trimmed Mean** and **Median** [42] filter out extreme updates to resist outliers. **FLTrust** [7] anchors updates to a trusted reference dataset. **FLAME** [33] incorporates client reputation into aggregation to downweight suspect clients.

**Provably robust FL:**  **FLCert** [11] trains multiple global models using subsets of clients, providing an ensemble-based lower bound on test accuracy even under strong attacks.

**Client detection:**  **FLDetector** [43] tracks consistency in model updates over time to identify malicious clients. **FedRecover** [10] recovers a clean global model by filtering out detected malicious updates, avoiding the need to retrain from scratch.

## B    Theoretical Analysis

FLForensics builds on Observations I and II (Section 4.3). In this section, we provide theoretical justification for these observations under a formal definition of poisoning attacks and several mild assumptions. While these assumptions may not always hold in practice, we empirically validate the effectiveness of FLForensics in Section 5.3.

### B.1    Setup and Assumptions

We first formalize poisoning attacks in FL. Malicious clients aim to make the global model predict the target label $y$ on any target input $x$, minimizing $\ell_{CE}(x, y; w)$, while benign clients aim to maintain accuracy on non-target inputs $x'$. This leads to the following assumptions (see Appendix B.2 for formal definitions):

- **Local Linearity**: Cross-entropy loss is approximately linear in a small neighborhood around the global model.
- **Behavioral Difference**:  Malicious clients tend to decrease $\ell_{CE}(x, y; w)$ but not $\ell_{CE}(x', y; w)$, while benign clients do the opposite.
- **Label-rich Advantage**: Category I benign clients, who have more data with target label, are more likely to behave like malicious clients on target inputs than Category II clients.

### B.2    Formal Definitions and Assumptions

**Definition 1** (Poisoning Attack to FL)**.**  *In a poisoning attack, malicious clients aim to poison the global model $w$ such that it predicts target label $y$ for any target input $x$, i.e., the loss $\ell_{CE}(x, y; w)$ is small; and benign clients aim to learn the global model such that it is accurate for non-target*

*true inputs, i.e., the loss $\ell_{CE}(x', y; w)$ is small. Therefore, in a training round, a malicious client's model update does not increase the loss $\ell_{CE}(x, y; w)$ for a target input, while a benign client's model update does not decrease such loss. On the contrary, a malicious client's model update does not decrease the loss $\ell_{CE}(x', y; w)$, while a benign client's model update does not increase such loss. Formally, for any check-point training round $t \in \Omega$, malicious client $i$, and benign client $j$, we have the following assumptions to characterize the training process:*

$$\ell_{CE}(x', y; w_t + g_t^{(j)}) \leq \ell_{CE}(x', y; w_t + g_t^{(i)}), \tag{6}$$

$$\ell_{CE}(x, y; w_t + g_t^{(j)}) \geq \ell_{CE}(x, y; w_t + g_t^{(i)}), \tag{7}$$

*where $w_t + g_t^{(i)}$ and $w_t + g_t^{(j)}$ respectively are the global models after training round $t$ if only the model updates of clients $i$ and $j$ were used to update the global model.*

**Assumption 1** (Local Linearity). *We assume the cross-entropy losses $\ell_{CE}(x', y; w_t)$ and $\ell_{CE}(x, y; w_t)$ are locally linear in the region around $w_t$. In particular, based on first-order Taylor expansion, we have the following:*

$$\ell_{CE}(x', y; w_t + \delta) = \ell_{CE}(x', y; w_t) + \nabla \ell_{CE}(x', y; w_t)^\top \delta,$$
$$\ell_{CE}(x, y; w_t + \delta) = \ell_{CE}(x, y; w_t) + \nabla \ell_{CE}(x, y; w_t)^\top \delta.$$

We note that the local linearity assumption was also used in the machine learning community [5, 41].

**Assumption 2** (Label-rich Advantage). *According to the definitions of Category I and Category II benign clients, a Category I benign client possesses a larger fraction of local training examples with the target label $y$ compared to a Category II benign client. Therefore, given any target input $x$ with target label $y$, we assume that the local model of a Category I benign client is more likely to predict $x$ as $y$ than that of a Category II benign client. Formally, for any check-point training round $t \in \Omega$, Category I benign client $j_1$, and Category II benign client $j_2$, we make the following assumption:*

$$\ell_{CE}(x, y; w_t + g_t^{(j_1)}) \leq \ell_{CE}(x, y; w_t + g_t^{(j_2)}), \tag{8}$$

*where $w_t + g_t^{(j_1)}$ is the local model of Category I benign client $j_1$ and $w_t + g_t^{(j_2)}$ is the local model of Category II benign client $j_2$ in the training round $t$.*

### B.3  Guarantee for Observation I

We show that under these assumptions, malicious clients and Category I benign clients have higher influence scores $s_i$ on target inputs than Category II benign clients. Theorem 1 and 2 together show that Observation I holds.

**Theorem 1.** *Suppose the server picks all clients in each check-point training round, i.e., $C_t = \{1, 2, \cdots, n\}$ for $t \in \Omega$, and FLForensics uses a true non-target input with target label $y$. Based on the poisoning attack definition and Assumption 1, we have that the influence score $s_i$ of a malicious client $i$ is no smaller than the influence score $s_j$ of a Category II benign client $j$. Concretely, we have $s_i \geq s_j$, where $s_i$ and $s_j$ are computed based on Equation 4.*

*Proof.* By setting $\delta = g_t^{(i)}$ in Assumption 1, we have the following for each check-point training round $t$:

$$\ell_{CE}(x, y; w_t + g_t^{(i)}) = \ell_{CE}(x, y; w_t) + \nabla \ell_{CE}(x, y; w_t)^\top g_t^{(i)}. \tag{9}$$

Similarly, by setting $\delta = g_t^{(j)}$ in Assumption 1, we have:

$$\ell_{CE}(x, y; w_t + g_t^{(j)}) = \ell_{CE}(x, y; w_t) + \nabla \ell_{CE}(x, y; w_t)^\top g_t^{(j)}. \tag{10}$$

By combining Equation 7, 9, and 10, we have:

$$\nabla \ell_{CE}(x, y; w_t)^\top g_t^{(i)} \leq \nabla \ell_{CE}(x, y; w_t)^\top g_t^{(j)}. \tag{11}$$

Since the learning rate $\alpha_t > 0$ and $C_t = \{1, 2, \cdots, n\}$ in each check-point training round, we have the following by summing over the check-point training rounds on both sides of Equation 11:

$$\sum_{t \in \Omega} \alpha_t \nabla \ell_{CE}(x, y; w_t)^\top g_t^{(i)} \leq \sum_{t \in \Omega} \alpha_t \nabla \ell_{CE}(x, y; w_t)^\top g_t^{(j)} \tag{12}$$

$$\Longleftrightarrow -s_i \leq -s_j. \tag{13}$$

Therefore, we have $s_i \geq s_j$, which completes the proof. $\qquad\square$

**Theorem 2.** *Suppose the server picks all clients in each check-point training round, i.e., $C_t = \{1, 2, \cdots, n\}$ for $t \in \Omega$, and FLForensics uses a true non-target input with target label $y$. Based on Assumption 1 and Assumption 2, we have that the influence score $s_{j_1}$ of a Category I benign client $j_1$ is no smaller than the influence score $s_{j_2}$ of a Category II benign client $j_2$. Specifically, we have $s_{j_1} \geq s_{j_2}$, where $s_{j_1}$ and $s_{j_2}$ are computed based on Equation 4.*

*Proof.* According to Assumption 2, we have:

$$\ell_{CE}(x, y; w_t + g_t^{(j_1)}) \leq \ell_{CE}(x, y; w_t + g_t^{(j_2)}). \tag{14}$$

By setting $\delta = g_t^{(j_1)}$ and $\delta = g_t^{(j_2)}$ in Assumption 1, we can get:

$$\ell_{CE}(x, y; w_t + g_t^{(j_1)}) = \ell_{CE}(x, y; w_t) + \nabla \ell_{CE}(x, y; w_t)^\top g_t^{(j_1)}, \tag{15}$$

$$\ell_{CE}(x, y; w_t + g_t^{(j_2)}) = \ell_{CE}(x, y; w_t) + \nabla \ell_{CE}(x, y; w_t)^\top g_t^{(j_2)}. \tag{16}$$

By combining Equation 14, 15, and 16, we have:

$$\nabla \ell_{CE}(x, y; w_t)^\top g_t^{(j_1)} \leq \nabla \ell_{CE}(x, y; w_t)^\top g_t^{(j_2)}. \tag{17}$$

Since the learning rate $\alpha_t > 0$ and $C_t = \{1, 2, \cdots, n\}$ in each check-point training round, we have the following by summing over the check-point training rounds on both sides of Equation 17:

$$\sum_{t \in \Omega} \alpha_t \nabla \ell_{CE}(x, y; w_t)^\top g_t^{(j_1)} \leq \sum_{t \in \Omega} \alpha_t \nabla \ell_{CE}(x, y; w_t)^\top g_t^{(j_2)} \tag{18}$$

$$\Longleftrightarrow -s_{j_1} \leq -s_{j_2}, \tag{19}$$

which gives $s_{j_1} \geq s_{j_2}$ and completes the proof. $\qquad\square$

### B.4 Guarantee for Observation II

We show that influence score gaps $s_i' - s_i$ are smaller for malicious clients than for benign ones.

**Theorem 3.** *Suppose the server picks all clients in each check-point training round, i.e., $C_t = \{1, 2, \cdots, n\}$ for $t \in \Omega$, and FLForensics uses a true non-target input with target label $y$. Based on the poisoning attack definition and Assumption 1, we have the influence score gap $s_i' - s_i$ of a malicious client $i$ is no larger than the influence score gap $s_j' - s_j$ of a benign client $j$. Formally, we have $s_i' - s_i \leq s_j' - s_j$, where $s_i$ and $s_j$ are computed based on Equation 4, while $s_i'$ and $s_j'$ are computed based on Equation 5.*

*Proof.* By setting $\delta = g_t^{(i)}$ in Assumption 1, we have the following for each check-point training round $t$:

$$\ell_{CE}(x', y; w_t + g_t^{(i)}) = \ell_{CE}(x', y; w_t) + \nabla \ell_{CE}(x', y; w_t)^\top g_t^{(i)}. \tag{20}$$

Similarly, by setting $\delta = g_t^{(j)}$ in Assumption 1, we have:

$$\ell_{CE}(x', y; w_t + g_t^{(j)}) = \ell_{CE}(x', y; w_t) + \nabla \ell_{CE}(x', y; w_t)^\top g_t^{(j)}. \tag{21}$$

By combining Equation 6, 20, and 21, we have:

$$\nabla \ell_{CE}(x', y; w_t)^\top g_t^{(i)} \geq \nabla \ell_{CE}(x', y; w_t)^\top g_t^{(j)}. \tag{22}$$

Since the learning rate $\alpha_t > 0$ and $C_t = \{1, 2, \cdots, n\}$ in each check-point training round, we have the following by summing over the check-point training rounds on both sides of Equation 22:

$$\sum_{t \in \Omega} \alpha_t \nabla \ell_{CE}(x', y; w_t)^\top g_t^{(i)} \geq \sum_{t \in \Omega} \alpha_t \nabla \ell_{CE}(x', y; w_t)^\top g_t^{(j)} \tag{23}$$

$$\Longleftrightarrow -s_i' \geq -s_j'. \tag{24}$$

By combining Equations 13 and 24, we have $s_i' - s_i \leq s_j' - s_j$, which completes the proof. $\qquad\square$

Table 3: Dataset statistics.

| Dataset | # Training | # Testing | # Classes |
|---|---|---|---|
| CIFAR-10 | 50,000 | 10,000 | 10 |
| Fashion-MNIST | 60,000 | 10,000 | 10 |
| MNIST | 60,000 | 10,000 | 10 |
| Sentiment140 | 72,491 | 358 | 2 |
| ImageNet-Fruits | 13,000 | 500 | 10 |

Table 4: CNN architecture for Fashion-MNIST and MNIST.

| Layer | Size |
|---|---|
| Input | $28 \times 28 \times 1$ |
| Convolution + ReLU | $3 \times 3 \times 30$ |
| Max Pooling | $2 \times 2$ |
| Convolution + ReLU | $3 \times 3 \times 50$ |
| Max Pooling | $2 \times 2$ |
| Fully Connected + ReLU | 100 |
| Softmax | 10 |

Table 5: Default parameter setting. Since ImageNet-Fruits only has 40 clients and 8 malicious clients, and "min_cluster_size" is set to 7, when performing clustering using HDBSCAN, we duplicate the influence scores of all clients before clustering, resulting in a total of 80 influence scores.

| Parameter | CIFAR-10 | Fashion-MNIST | MNIST | Sentiment140 | ImageNet-Fruits |
|---|---|---|---|---|---|
| # clients | | | 100 | | 40 |
| # malicious clients | | | 20 | | 8 |
| # rounds | 1500 | 2000 | 2000 | 1500 | 1000 |
| # local training epochs | | | 1 | | |
| Batch size | 64 | 32 | 32 | 32 | 64 |
| Learning rate | $1 \times 10^{-2}$ | $6 \times 10^{-3}$ | $3 \times 10^{-4}$ | $1 \times 10^{-1}$ (decay at the 800th epoch with factor 0.5) | $1 \times 10^{-2}$ |
| # check points | 150 | 200 | 200 | 150 | 100 |
| min_cluster_size | | | 7 | | |

**Corollary 1.** *Given a malicious client $i$ and a benign client $j$, if the influence scores $s_i > 0$ and $s_j > 0$, then the influence score ratios satisfy: $\frac{s'_i}{s_i} \leq \frac{s'_j}{s_j}$.*

*Proof.* From Equation 13 and Equation 24, we have $s'_i \leq s'_j$ and $s_i \geq s_j$. Since $s_i > 0$ and $s_j > 0$, we have:

$$\frac{s'_i}{s_i} \leq \frac{s'_i}{s_j} \leq \frac{s'_j}{s_j}, \tag{25}$$

which completes the proof. □

This directly implies the following:

**Corollary 2.** *If $s_i, s_j > 0$, then $\frac{s'_i}{s_i} \leq \frac{s'_j}{s_j}$.*

**Remark.** *This explains why FLForensics can use the ratio $s'_i/s_i$ to distinguish malicious clients from Category I benign clients when both fall into the same high-$s_i$ cluster.*

## C   Dataset Description

We conduct our experiments using five diverse benchmark datasets: four image datasets (CIFAR-10, Fashion-MNIST, MNIST, and ImageNet-Fruits) and one text dataset (Sentiment140). Table 3 summarizes their key statistics.

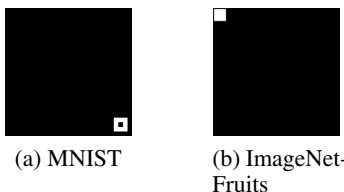

(a) MNIST    (b) ImageNet-Fruits

Figure 5: Triggers in MNIST and ImageNet-Fruits datasets.

**CIFAR-10 [29].** This is a commonly employed dataset in image classification task, comprising 50,000 training examples and 10,000 testing examples. Each input is a 3-channel color image of 32×32 pixels in size and belongs to one of ten classes. *Model: ResNet-20 [26], implemented using MXNet. License: Apache License 2.0. Dataset License: MIT License.* `https://www.cs.toronto.edu/~kriz/cifar.html`

**Fashion-MNIST [38].** This dataset consists of 70,000 grayscale images of fashion items, divided into 60,000 training examples and 10,000 testing examples. Each input is of 28×28 pixels in size and belongs to one of ten classes. *Model: A custom CNN architecture implemented using MXNet (Table 4). Dataset License: MIT License.* `https://github.com/zalandoresearch/fashion-mnist`

**MNIST [30].** Like Fashion-MNIST, MNIST also contains 70,000 1-channel grayscale images, split into 60,000 training examples and 10,000 testing examples. Each input is a 28×28 pixel image of a handwritten digit. The dataset includes ten classes with each class corresponding to a digit from 0 to 9. *Model: Same CNN architecture as Fashion-MNIST, implemented using MXNet. Dataset License: Creative Commons Attribution-Share Alike 3.0.* `http://yann.lecun.com/exdb/mnist/`

**Sentiment140 [23].** This is a two-class text classification dataset for sentiment analysis. The dataset is collected from Twitter users. In our experiments, we adopt users with at least 50 tweets, which results in 927 users. Each user has a pre-defined set of training and testing tweets. For our considered users, we have 72,491 training tweets and 358 testing tweets in total. *Model: LSTM [27], implemented using MXNet. License: Apache License 2.0. Dataset License: Other (academic use only).* `https://huggingface.co/datasets/stanfordnlp/sentiment140`

**ImageNet-Fruits [12].** This is an image classification dataset comprising 128×128 pixel color images. It represents a subset of the larger ImageNet-1k [15] dataset, specifically curated to include ten fruit categories. *Model: ResNet-50 [26], implemented using MXNet. License: Apache License 2.0. Dataset License: ImageNet terms (non-commercial research only).* `https://image-net.org/download`

## D  Poisoning Attack Description

**Scaling [3].** Following [3], malicious clients duplicate their local data, embed a trigger, relabel these copies with the *target label*, and train on the mix of original and duplicated samples. Furthermore, malicious clients scale their updates by a factor $\gamma$ before sending them to the server. We set $\gamma = 1$ by default, since it is stealthy yet still effective. Triggers follow [3] for CIFAR-10 and [24] for Fashion-MNIST; those for MNIST and ImageNet-Fruits appear in Fig. 5a and Fig. 5b. For Sentiment140 we insert the phrase 'debug FLpoisoning' in place of two consecutive words. Target labels are 2 for CIFAR-10, 0 for Fashion-MNIST and MNIST, 'negative' for Sentiment140, and 2 for ImageNet-Fruits.

**A little is enough (ALIE) [4].** The attacker in ALIE attack uses the same strategy as that in Scaling attack to embed the triggers into duplicated local training inputs and set their labels as target labels on the malicious clients. However, instead of scaling the model updates, the malicious clients carefully craft their model updates via solving an optimization problem.

**Edge [37].** The attacker in Edge attack injects some training examples (called *edge-case examples*) labeled as the target label into the malicious clients' local training data, which are from a distribution different from that of the learning task's overall training data. Each malicious client trains its local model using the original local training examples and the edge-case ones following the FL

algorithm. In our experiments, for CIFAR-10 and Sentiment140 datasets, we use the edge-case examples respectively designed for CIFAR-10 and Sentiment140 datasets in [37]. For the Fashion-MNIST and MNIST datasets, we use the edge-case examples designed for EMNIST dataset in [37]. For ImageNet-Fruits dataset, we use unripe banana images from [34] as edge-case examples and label them as 'cucumber'. The target inputs are from the same dataset as the edge-case examples.

## E    Extending GAS to FL

Since the total number of training examples possessed by all clients is much larger than the number of clients, when we detect malicious training examples, we use HDBSCAN to divide the examples into a big cluster (size is at least half of the whole training dataset size) and *outliers* with respect to their influence scores by setting "*min_cluster_size*" as $|D|/2 + 1$, where $|D|$ is the whole training dataset size. Unlike FLForensics, we adopt the Euclidean distance metric for HDBSCAN since GAS only has one-dimensional influence score, for which scaling is meaningless. We then use the outliers to determine the threshold since the big cluster corresponds to the majority clean training examples. Specifically, HDBSCAN outputs a *confidence level* (a number between 0 and 1) for each outlier, which indicates the confidence HDBSCAN has at predicting an input as outlier. We adopt a confidence level of 95%, which is widely used in statistics. Specifically, we treat the outliers whose confidence levels are at least 95% and whose influence scores are positive as "true" outliers. Moreover, we set the smallest influence score of such true outliers as the threshold.

## F    Simulating Non-iid Setting in FL

Following previous works [7, 16], to simulate the non-iid data distribution across clients, we randomly partition all clients into $C$ groups, where $C$ is the number of classes. We then assign each training example with label $y$ to the clients in one of these $C$ groups with a probability. In particular, a training example with label $y$ is assigned to clients in group $y$ with a probability of $\rho$, and to clients in any other groups with an equal probability of $\frac{1-\rho}{C-1}$, where $\rho \in [0.1, 1.0]$. Within the same group, the training example is uniformly distributed among the clients. Therefore, $\rho$ controls the degree of non-iid. When $\rho = 0.1$, the local training data follows an iid distribution in our datasets; otherwise, the clients' local training data is non-iid. A higher value $\rho$ implies a higher degree of non-iid.

## G    Details of Other Experiments

**Using true input as non-target input:**  In our experiments, we use a random input (e.g., a random image) as a non-target input in FLForensics. When a true input with the target label is available, the server can also use it as the non-target input. Table 6 compares the results when FLForensics uses a random input or true input as the non-target input on the five datasets and three attacks. We find that random inputs and true inputs achieve comparable results in most cases, except several cases, for which true inputs achieve slightly lower FPRs. These results indicate that if a true input with the target label from the learning task's data distribution is available, the server can use it as the non-target input.

**Forensics for clean-label targeted attacks:**  We focus on *dirty-label* attacks, where poisoned samples are relabeled to the *target label*. However, we find that FLForensics also works well for *clean-label* attacks, where labels remain unchanged, since it relies on clients' model updates rather than training data. Once those updates are backdoored, FLForensics can still trace back. We test this using the clean-label attack [35] on CIFAR-10 with FedAvg, training 'cat' images to resemble 'dog' features while keeping their original labels. Under default settings, FLForensics achieves DACC=0.95, FPR=0.06, and FNR=0.0, confirming its effectiveness.

**Extending FLForensics to centralized learning:**  FLForensics can also be extended to centralized learning by treating each training example as a client.' In this setting, $w_t$ in Equation 4 is the model at the $t$th mini-batch, and $g_t^{(i)}$ is the gradient of the loss of $w_t$ on the $i$th example. Table 7 shows results on the dataset sampled from CIFAR-10, where we inject triggers (as in our FL setup) into 10% of training data with target label 1. FLForensics-True denotes the use of a true input as the non-target input. Class distributions are shown in Figure 6. When the dataset is imbalanced, which is similar

Table 6: Results of FLForensics when using a random or true input as a non-target input.

| Dataset | Non-target input | Scaling attack | | | ALIE attack | | | Edge attack | | |
|---|---|---|---|---|---|---|---|---|---|---|
| | | DACC | FPR | FNR | DACC | FPR | FNR | DACC | FPR | FNR |
| CIFAR-10 | Random | 1.000 | 0.000 | 0.000 | 1.000 | 0.000 | 0.000 | 0.980 | 0.000 | 0.100 |
| | True | 1.000 | 0.000 | 0.000 | 1.000 | 0.000 | 0.000 | 0.970 | 0.013 | 0.100 |
| Fashion-MNIST | Random | 1.000 | 0.000 | 0.000 | 1.000 | 0.000 | 0.000 | 1.000 | 0.000 | 0.000 |
| | True | 1.000 | 0.000 | 0.000 | 1.000 | 0.000 | 0.000 | 1.000 | 0.000 | 0.000 |
| MNIST | Random | 1.000 | 0.000 | 0.000 | 1.000 | 0.000 | 0.000 | 1.000 | 0.000 | 0.000 |
| | True | 1.000 | 0.000 | 0.000 | 1.000 | 0.000 | 0.000 | 1.000 | 0.000 | 0.000 |
| Sentiment140 | Random | 0.980 | 0.025 | 0.000 | 1.000 | 0.000 | 0.000 | 0.970 | 0.038 | 0.000 |
| | True | 0.990 | 0.013 | 0.000 | 1.000 | 0.000 | 0.000 | 0.990 | 0.013 | 0.000 |
| ImageNet-Fruits | Random | 1.000 | 0.000 | 0.000 | 1.000 | 0.000 | 0.000 | 0.950 | 0.031 | 0.125 |
| | True | 1.000 | 0.000 | 0.000 | 1.000 | 0.000 | 0.000 | 0.950 | 0.031 | 0.125 |

Figure 6: Number of training examples for each class, where class distribution is roughly a power-law. We sample the unbalanced dataset from CIFAR-10. We have 20,431 training examples in total.

Table 7: Results for centralized learning. The class distributions are shown in Figure 6.

| Method | DACC | FPR | FNR |
|---|---|---|---|
| PF | 0.831 | 0.189 | 0.000 |
| FLForensics-G | 0.828 | 0.191 | 0.000 |
| FLForensics-A | 0.826 | 0.194 | 0.000 |
| FLForensics | 0.992 | 0.008 | 0.005 |
| FLForensics-True | 0.998 | 0.002 | 0.000 |

Table 8: Storage overhead of FLForensics.

| Dataset | # check points | # clients | Storage overhead (GB) |
|---|---|---|---|
| CIFAR-10 | 150 | 100 | 15.22 |
| Fashion-MNIST | 200 | 100 | 10.43 |
| MNIST | 200 | 100 | 10.43 |
| Sentiment140 | 150 | 100 | 3.02 |
| ImageNet-Fruits | 100 | 40 | 22.20 |

to non-IID data in FL, and the target label belongs to a majority class, FLForensics significantly outperforms existing poison-forensics methods.

# H   Other Ablation Studies

In this Section, we provide other ablation studies for FLForensics.

**Impact of number of check points and storage overhead:** Figure 7a shows the impact of the number of check points on FLForensics. We observe a trade-off between storage overhead and poison-forensics performance. In particular, when a server saves more check points, which incurs more storage overhead, the server can more accurately detect the malicious clients. We also note

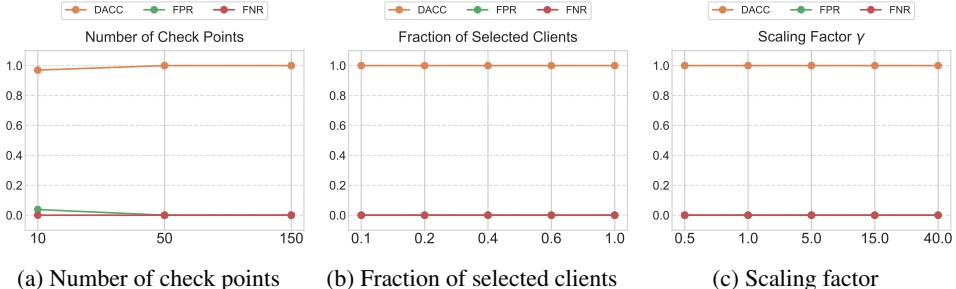

(a) Number of check points    (b) Fraction of selected clients    (c) Scaling factor

Figure 7: Results of other ablation studies for FLForensics.

that the storage overhead is acceptable for a powerful server to achieve good poison-forensics performance. Table 8 shows the storage overhead for different datasets in our default settings. For instance, saving 150 check points for CIFAR-10 requires 15.22GB storage, which is acceptable for a powerful server like a data center.

**Impact of fraction of selected clients:** Based on Figure 7b, FLForensics works well even if the server selects a small fraction of clients in each training round. The reason is that once a malicious client appears in multiple check-point training rounds, FLForensics can accumulate its influences to distinguish it with benign clients. Note that in this experiment, we save all training rounds as check points so a malicious client appears in multiple of them.

**Impact of scaling factor $\gamma$:** Based on the results in Figure 7c, we observe that FLForensics works well for a wide range of scaling factors used by the Scaling attack. The reason is that once the Scaling attack is effective, FLForensics can identify the malicious clients. Previous work [3] shows that the higher the scaling factor, the more effective the attack, but it also makes the attack more susceptible to detection. We can observe that our FLForensics can detect all the malicious clients even when the scaling factor is close to 1. In such cases, methods (e.g., FLDetector) that rely on the magnitudes of clients' model updates to detect malicious clients often fail.

# I Limitation

**Forensics for untargeted poisoning attacks:** This work focuses on forensics for *targeted* poisoning attacks. A promising direction for future work is extending FLForensics to handle *untargeted* attacks [16], where the poisoned model misclassifies many clean inputs, resulting in low test accuracy. Our current method is less effective in this setting, as the misclassified input in an untargeted poisoning attack is also a non-target input.

# J Broader Impact

Our work proposes FLForensics, the first method to trace back malicious clients in poisoning attacks to FL. It addresses a critical gap in existing defenses, which primarily focus on preventing attacks during training. FLForensics provides a complementary line of defense by enabling post-deployment forensics, which is especially important when training-phase defenses fail.

This contribution has positive societal impact, as it helps increase accountability in collaborative learning systems deployed in sensitive domains such as healthcare, finance, and mobile platforms. By identifying malicious participants after attack-induced misclassifications, FLForensics promotes the development of more trustworthy and robust federated systems.

Table 9: Test accuracy (TACC) and attack success rate (ASR) of different FL aggregation rules under different attacks. For the Trim aggregation rule, the trim parameter is set to the number of malicious clients. The server in FLTrust holds a small and clean root dataset. In our experiments, the size of the root dataset is set to 50, and the root dataset is drawn from the same distribution as that of the learning task's overall training data. For the FLAME aggregation rule, we use the same parameters as in [33]. We do not show ASR when there is no attack (i.e., "—") because different attacks use different triggers.

(a) CIFAR-10

| Attack | FedAvg | | Trim | | Median | | FLTrust | | FLAME | |
|---|---|---|---|---|---|---|---|---|---|---|
| | TACC | ASR | TACC | ASR | TACC | ASR | TACC | ASR | TACC | ASR |
| No attack | 0.837 | — | 0.769 | — | 0.755 | — | 0.811 | — | 0.774 | — |
| Scaling attack | 0.831 | 0.682 | 0.777 | 0.950 | 0.780 | 0.930 | 0.816 | 0.642 | 0.776 | 0.644 |
| ALIE attack | 0.843 | 0.956 | 0.814 | 0.754 | 0.809 | 0.980 | 0.806 | 0.968 | 0.780 | 0.958 |
| Edge attack | 0.819 | 0.174 | 0.761 | 0.337 | 0.762 | 0.352 | 0.794 | 0.056 | 0.789 | 0.087 |

(b) Fashion-MNIST

| Attack | FedAvg | | Trim | | Median | | FLTrust | | FLAME | |
|---|---|---|---|---|---|---|---|---|---|---|
| | TACC | ASR | TACC | ASR | TACC | ASR | TACC | ASR | TACC | ASR |
| No attack | 0.900 | — | 0.856 | — | 0.864 | — | 0.880 | — | 0.887 | — |
| Scaling attack | 0.887 | 0.953 | 0.870 | 0.892 | 0.841 | 0.043 | 0.874 | 0.037 | 0.890 | 0.024 |
| ALIE attack | 0.889 | 0.941 | 0.809 | 0.113 | 0.764 | 0.040 | 0.876 | 0.038 | 0.886 | 0.020 |
| Edge attack | 0.886 | 0.990 | 0.862 | 1.000 | 0.856 | 1.000 | 0.861 | 0.990 | 0.883 | 0.990 |

(c) MNIST

| Attack | FedAvg | | Trim | | Median | | FLTrust | | FLAME | |
|---|---|---|---|---|---|---|---|---|---|---|
| | TACC | ASR | TACC | ASR | TACC | ASR | TACC | ASR | TACC | ASR |
| No attack | 0.960 | — | 0.948 | — | 0.943 | — | 0.926 | — | 0.948 | — |
| Scaling attack | 0.958 | 0.950 | 0.921 | 0.013 | 0.936 | 0.010 | 0.926 | 0.006 | 0.954 | 0.005 |
| ALIE attack | 0.958 | 0.944 | 0.788 | 0.045 | 0.929 | 0.013 | 0.927 | 0.007 | 0.953 | 0.005 |
| Edge attack | 0.953 | 0.990 | 0.938 | 0.980 | 0.939 | 0.960 | 0.919 | 0.580 | 0.953 | 0.070 |

(d) SENT140

| Attack | FedAvg | | Trim | | Median | | FLTrust | | FLAME | |
|---|---|---|---|---|---|---|---|---|---|---|
| | TACC | ASR | TACC | ASR | TACC | ASR | TACC | ASR | TACC | ASR |
| No attack | 0.660 | — | 0.684 | — | 0.673 | — | 0.494 | — | 0.589 | — |
| Scaling attack | 0.659 | 0.995 | 0.616 | 0.985 | 0.645 | 1.000 | 0.494 | 1.000 | 0.592 | 0.232 |
| ALIE attack | 0.687 | 1.000 | 0.654 | 1.000 | 0.531 | 0.801 | 0.494 | 1.000 | 0.651 | 0.122 |
| Edge attack | 0.564 | 0.600 | 0.567 | 0.683 | 0.609 | 0.858 | 0.494 | 1.000 | 0.581 | 0.175 |

(e) ImageNet-fruits

| Attack | FedAvg | | Trim | | Median | | FLTrust | | FLAME | |
|---|---|---|---|---|---|---|---|---|---|---|
| | TACC | ASR | TACC | ASR | TACC | ASR | TACC | ASR | TACC | ASR |
| No attack | 0.517 | — | 0.537 | — | 0.509 | — | 0.480 | — | 0.492 | — |
| Scaling attack | 0.494 | 0.881 | 0.465 | 0.749 | 0.467 | 0.842 | 0.473 | 0.108 | 0.494 | 0.102 |
| ALIE attack | 0.488 | 1.000 | 0.486 | 0.113 | 0.469 | 0.132 | 0.482 | 0.115 | 0.502 | 0.952 |
| Edge attack | 0.514 | 0.283 | 0.529 | 0.279 | 0.502 | 0.219 | 0.490 | 0.377 | 0.470 | 0.465 |

Table 10: Results of PF and FLForensics-G when they use the clients' model updates.

| Method | Scaling attack | | | ALIE attack | | | Edge attack | | |
|---|---|---|---|---|---|---|---|---|---|
| | DACC | FPR | FNR | DACC | FPR | FNR | DACC | FPR | FNR |
| PF | 0.900 | 0.125 | 0.000 | 0.900 | 0.125 | 0.000 | 0.920 | 0.100 | 0.000 |
| FLForensics-G | 0.850 | 0.150 | 0.150 | 0.880 | 0.150 | 0.000 | 0.880 | 0.138 | 0.050 |

