# OpenReview forum: "Tracing Back the Malicious Clients in Poisoning Attacks to Federated Learning"
_NeurIPS.cc/2025/Conference — NeurIPS 2025 poster_

### Official Review · Reviewer_p7T5 · 2025-06-08

**Clarity:** 4
**Significance:** 3
**Originality:** 3
**Rating:** 5
**Confidence:** 4

**Summary:**

The paper presents a framework called FLForensics that allows to reliably identify malicious clients that ran targeted poisoning attacks - even if they managed to slip through poisoning defences such as robust aggregation. For this, the aggregation server keeps a subset of model snapshots and submitted client updates. When an attack is detected for a deployed model through unexpected misclassifications, this repository allows to trace back the malicious clients by computing influence scores for target and non-target inputs. Using an unsupervised clustering algorithm on these two-dimensional scores then allows to reliably classify malicious clients even in settings with non-iid data. The authors implement FLForensics and test its effectiveness on a range of datasets and models, data distributions, poisoning attacks, and (robust) aggregation rules. This demonstrates the effectiveness through a very high detection accuracy, also in comparison to prior works that have been adapted for the setting.

**Questions:**

Only two questions came to mind while reading the paper:
- As an optimization the aggregation server only stores a subset (10%?) of all intermediate states. However, it would be interesting to know if this impacts the potential for recovery from the attack. If all interim models and client updates would be available, would it be possible to compute a "fixed" model that tries to exclude/reverse the effect of the malicious clients without having to completely re-run the training process?
- If attackers are aware that FLForensics is deployed, could they adapt the attacks in a way that their non-target influence score would be less suspicious?

**Ethical Concerns:**

["NO or VERY MINOR ethics concerns only"]

**Final Justification:**

Thank you for following up on my two suggestions/questions in the rebuttal. There are no concerns from my side regarding this paper, so retain my positive rating.

**Limitations:**

yes

**Quality:**

3

**Strengths And Weaknesses:**

The paper is very well written and accessible. It provides a clear motivation for studying poisoning detection/tracing even in the presence of robust aggregation. The assumptions required for FLForensics to function properly are clearly articulated and appear very reasonable. The proposed approach is accompanied by a thorough formal analysis. The evaluation results are impressive in terms of effectiveness and consider a wide range of datasets and models, aggregation rules, poisoning attacks, data distributions, and most importantly also adapt prior works for this setting for a fair comparison. I was not able to spot weaknesses in the paper. Therefore, I recommend acceptance.

---

> ### Author Rebuttal · Authors · 2025-07-30
>
> We sincerely appreciate your recognition of our work! Here are our responses to your questions:
>
> Q1: **Recovery from the attack**: This is a very valuable suggestion. Indeed, if we save the checkpoint of each training round, including both the clients' updates and the global models, we can retain only the updates from benign clients, remove those identified as malicious by FLForensics, and directly perform aggregation, thus avoiding a complete re-run. On the other hand, if only partial intermediate states are saved, we may have to resume training from a specific intermediate round, which can still help mitigate the impact of backdoor attacks. Additionally, techniques such as unlearning [1] or recovery-based methods [2] can be used to further exclude the influence of malicious clients.
>
> [1] Zeng, Yi, et al. "Adversarial unlearning of backdoors via implicit hypergradient." ICLR, 2022.
>
> [2] Cao, Xiaoyu, et al. "Fedrecover: Recovering from poisoning attacks in federated learning using historical information." 2023 IEEE Symposium on Security and Privacy. IEEE, 2023.
>
> Q2: **Adapt the attacks**: Since FLForensics only stores a subset of all intermediate states, malicious clients can evade the defense by reducing the frequency or probability of their attacks rather than launching them in every round. The top two subfigures in Figure 4 illustrate the results of such adaptive attacks. We observe that FLForensics is still able to accurately identify the malicious clients. However, when the attack becomes less effective, for example, with an attack frequency of 10 or an attack probability of 0.2, the influence scores of malicious clients become less suspicious, leading to a slight increase in the FPR and FNR of FLForensics. This highlights a trade-off between attack effectiveness and stealthiness.

---

> ### Comment · Reviewer_p7T5 · 2025-08-05
> **Response to Rebuttal**
>
> Thank you for following up on my two suggestions/questions in the rebuttal. There are no concerns from my side regarding this paper, so retain my positive rating.

---

> > ### Author Response · Authors · 2025-08-05
> >
> > Thank you for your thoughtful review and for taking the time to consider our responses. We appreciate your feedback and your continued engagement with our work.

---

### Official Review · Reviewer_7o1F · 2025-06-30

**Clarity:** 2
**Significance:** 1
**Originality:** 2
**Rating:** 2
**Confidence:** 5

**Summary:**

The paper proposes FLForensics, a method to identify malicious clients in federated learning (FL) after a poisoning attack has taken place. The approach assumes access to individual client updates and relies on storing model updates at selected checkpoints during training. Once a misclassified target input is identified, the server computes per-client influence scores and uses clustering to distinguish malicious clients from benign ones. The authors provide a theoretical analysis under simplifying assumptions and evaluate the method on several datasets and poisoning attacks. The entire framework is built on vanilla FL, where no privacy mechanism such as secure aggregation is used.

**Questions:**

I apologize if my evaluation comes across as harsh, but I do not believe there are any revisions that could bring this paper to an acceptable standard for NeurIPS.

**Ethical Concerns:**

["NO or VERY MINOR ethics concerns only"]

**Final Justification:**

The main (significant) weakness I pointed our remains:  The approach proposed by the authors requires accessing the individual models, which significantly simplifies the problem.

While the paper has some merits, I do not think the work meets the standards for a top-tier conference like NeurIPS.

**Limitations:**

A significant limitation of the paper is its reliance on vanilla federated learning, which is known to offer no privacy protection and is rarely used in practice without secure aggregation. More critically, the authors' claim that "we propose the first poison-forensics method called FLForensics to trace back malicious clients in FL" is not true. There is prior work that addresses malicious client identification in federated learning, including settings with privacy constraints.

**Quality:**

1

**Strengths And Weaknesses:**

Strengths:  The topic addressed by the paper, i.e., identifying malicious clients in federated learning (FL), is both important and timely. However, the paper suffers from several significant weaknesses that limit its overall contribution and practical relevance.

Weaknesses:

1. A fundamental limitation of the paper is that it assumes a vanilla FL setup in which clients send their raw local model updates to the server without any privacy-preserving mechanism. This setup is largely unrealistic in real-world applications (note that it does not provide privacy), where secure aggregation is the de facto standard. In such settings, the server only observes the aggregated model update and has no access to individual client contributions. The proposed FLForensics framework critically relies on access to per-client updates, rendering it inapplicable with secure aggregation. Consequently, the practical relevance of the proposed method is very limited.

Moreover, there is a growing body of literature that addresses poisoning attacks and even malicious client identification under secure aggregation, which the authors do not sufficiently acknowledge. For instance, methods like FedGT (Xemrishi et al., TIFS 2025) directly tackle malicious client detection in a secure aggregation setting, a significantly more challenging and realistic problem.

2. The paper makes overstated and inaccurate claims, such as being "the first poison-forensics method for FL to trace back malicious clients." This is simply incorrect. There is a substantial body of prior work that addresses malicious client identification in federated learning, both in vanilla FL and under secure aggregation. Examples include:

Li et al. ``Learning to detect malicious clients for robust federated learning,'' 2020

Xemrishi et al. ``FedGT: Identification of Malicious Clients in Federated Learning with Secure Aggregation,'' TIFS, 2025.

Bellafqira et al. ``FedCAM - Identifying Malicious Models in Federated Learning Environments Conditionally to Their Activation Maps''

Mallah, ``Untargeted poisoning attack detection in federated learning via behavior attestationAl'', IEEE Access 2023

These works propose a variety of methods for detecting or identifying malicious clients during or after training, and some even operate under secure aggregation, which makes the problem more realistic and challenging. The paper under review fails to acknowledge this body of literature, which undermines its novelty claims and weakens its contribution. Overall, the claim of being the "first" to address this problem is both misleading and unjustified.

3. The threat model is not clearly defined. While the authors state that they focus on targeted poisoning attacks, the experimental evaluation is based on attacks such as Scaling [3], ALIE [4], and Edge-case attacks [30], which are not targeted attacks in the sense commonly used in the literature. This inconsistency weakens the validity of the empirical evaluation and raises concerns about the alignment between the stated objectives and the experiments.

4. The writing quality of the paper is poor, with numerous inconsistencies and lapses in clarity, particularly in the definition of the threat model and in some methodological descriptions. Parts of the text appear hastily written or potentially AI-generated, giving the impression that the submission was rushed. Regardless of authorship, the paper does not meet the minimum presentation standards expected for a NeurIPS submission, which makes it difficult to thoroughly evaluate the technical contributions.

5. The results section is weak.

---

> ### Author Rebuttal · Authors · 2025-07-30
>
> Thank you for the insightful comments. Here are our responses to your comments.
>
> W1. **Assumes a vanilla FL**:
>
> 1). Following many prior works on federated learning defenses (e.g., “Learning to detect...”, FedCAM, and “Untargeted poisoning... ” mentioned by the reviewer), we adopt the commonly used assumption that the server is honest and directly receives model updates from clients.
>
> 2). If a trusted third party is available to assist with secure aggregation, it can also be leveraged to perform the forensic traceback procedure, which is orthogonal to our proposed method and can complement it.
>
> 3). We appreciate the reference to FedGT. As this line of work suggests, one possible way to ensure privacy while still enabling malicious client detection is to operate on aggregated group updates rather than individual ones. We believe our traceback approach could potentially be extended to such a group-based secure aggregation setting. In particular, instead of computing influence scores over individual client updates, we can compute group-level influence scores over aggregated updates, where each group consists of a small subset of clients. By designing overlapping groups and aggregating their updates, our traceback mechanism can still identify malicious clients using group-level influence patterns, similar in spirit to approaches like FedGT. We view this as a promising direction for future work.
>
> 4). Finally, we note that in many practical FL deployments (especially in cross-silo settings), secure aggregation is not yet widely adopted due to its added communication and system complexity. Thus, our work remains applicable to a broad range of real-world scenarios.
>
> W2: **Overstated and inaccurate claims**: Our work is **poison-forensics**, meaning that after FL training is completed (even after the model has been deployed), if a user reports a misclassification event, FLForensics can trace back the malicious clients who participated in training. The papers mentioned by the reviewer have a different goal from ours. All four papers are **FL aggregation rules**, not post-training poison-forensics. Specifically:
>
> “Learning to detect...” states: “After obtaining the spectral anomaly detection model, we apply it in every round of the FL model training to detect malicious client updates.”
>
> “FedGT...” is also an aggregation rule: “a novel framework for identifying malicious clients in federated learning with secure aggregation”
>
> “FedCAM” also states: “During each FL training round, the server calculates the geomed normalized AMs for each client-updated model using the trigger set”
>
> "Untargeted poisoning..." states: “attestedFL removes the local models that are unreliable before computing the global model in each iteration of federated learning”
>
> The above evidences show that all of these works perform defense during FL training, and can thus be considered robust aggregation rules. In contrast, our work is post-training poison-forensics. Therefore, we believe our claim, "the first poison-forensics method for FL to trace back malicious clients.", is correct.
>
> W3: **Attacks are not commonly used in the literature**: Actually, the three attack papers we evaluate are all highly impactful, commonly used targeted attacks. Specifically, Scaling (published at AISTATS 2020) has around 2800 citations on Google Scholar, ALIE (NeurIPS 2019) has over 700 citations, and Edge-case (NeurIPS 2020) has over 800 citations.
>
> W4: **Writing quality**: We would greatly appreciate it if the reviewer could provide specific suggestions (for example, where in the paper you observed inconsistencies or lapses), so that we can revise accordingly. However, we would like to clarify that our paper is **not AI-generated**.
>
> W5: **Results section is weak**: We would appreciate it if the reviewer can point out which specific parts of our results are considered weak, so we will carefully consider and address the feedback.

---

> > ### Comment · Reviewer_7o1F · 2025-08-03
> >
> > The authors are mistaken in their characterization of FedGT. It is not a robust aggregation rule. As is evident from the FedGT paper itself (see its introduction and methodology), the goal of FedGT is to identify malicious clients under secure aggregation constraints, not to aggregate in a robust manner during training. Once malicious clients are identified, the training process is restarted without them. This is fundamentally different from robust aggregation. I encourage the authors to revisit the FedGT paper more carefully.
> >
> > My main concern remains unaddressed. The authors' approach assumes a vanilla FL setup in which the server has direct access to individual client model updates. This assumption violates the core principle of federated learning: client privacy. In fact, once the server can inspect individual updates, identifying malicious clients becomes significantly easier, and the forensic challenge the authors propose to solve is arguably less meaningful.  In this setting, it is not surprising that the reported results are strong.
> >
> > Without addressing more realistic scenarios that preserve privacy, such as those incorporating differential privacy and/or secure aggregation, the practical impact of the proposed method remains highly limited. Federated learning systems that do not enforce such protections are simply not going to be deployed in privacy-sensitive applications.
> >
> > Accordingly, I am unable to raise my score at this time.

---

> ### Author Response · Authors · 2025-08-04
>
> Regarding FedGT, we appreciate the reviewer's clarification. Based on the reviewer's description, we believe FedGT is more similar to FLDetector [1], which also detects and removes malicious clients during the training process. In our paper, we discussed the limitations of FLDetector and its differences from our approach in the related work section (line 93 and 262 in the main paper).
>
> Overall, FedGT, FLDetector, and secure aggregation rules are all training-phase defenses, whereas our paper focuses on post-training forensics. Therefore, FLForensics is fundamentally different from FedGT.
>
> For realistic scenarios that preserve privacy, we add noise to all clients' model updates based on the widely adopted differential privacy mechanism [2]. Specifically, we use the following formula to add noise:
> $$
> \bar{g}_ t(x_ i) \leftarrow g_ t(x_ i) / \max\left(1, \frac{||g_ t(x_ i)||_ 2}{C}\right), \
> \tilde{g}_ t \leftarrow \frac{1}{L} \left( \sum_ i \bar{g}_ t(x_ i) + \mathcal{N}(0, \sigma^2 C^2 \mathbf{I})\right)
> $$
>  We experiment with different noise levels (i.e., different C and $\sigma$), and the results are as follows:
>
> |$C$|$\sigma$|TACC|ASR|DACC/FPR/FNR|
> |-|-|-|-|-|
> |10.0|0.05|0.483|0.950|0.980/0.000/0.100|
> |10.0|0.01|0.675|0.953|1.000/0.000/0.000|
> |10.0|0.001|0.801|0.765|1.000/0.000/0.000|
> |1.0|0.01|0.602|0.952|1.000/0.000/0.000|
>
>  The results demonstrate that FLForensics can successfully trace back under varying levels of privacy.
>
> [1]. Zhang, Zaixi, et al. "Fldetector: Defending federated learning against model poisoning attacks via detecting malicious clients." Proceedings of the 28th ACM SIGKDD conference on knowledge discovery and data mining. 2022.
>
> [2]. Abadi, Martin, et al. "Deep learning with differential privacy." Proceedings of the 2016 ACM SIGSAC conference on computer and communications security. 2016.

---

> > ### Comment · Reviewer_7o1F · 2025-08-07
> >
> > Thank you for the additional clarifications.
> >
> > While I acknowledge that the paper has some merit, I still believe that my main concerns remain unaddressed, particularly the assumption of access to individual models, which significantly simplifies the problem. As such, I do not think the work meets the standards for a top-tier conference like NeurIPS.

---

### Official Review · Reviewer_Y9xs · 2025-07-02

**Clarity:** 3
**Significance:** 2
**Originality:** 2
**Rating:** 4
**Confidence:** 4

**Summary:**

This paper proposes a mechanism to detect malicious clients in an FL protocol in the aftermath of data poisoning. By examining the influence score of model updates to the misclassified instance, the protocol is able to detect malicious clients. The forensic detection is effective across multiple datasets, aggregation rules and attack methods.

**Questions:**

1. The attack success rate for Scaling attack against FLTrust on CIFAR10 (Table 6a) is very different from the one in Table III d) in the original FLTrust paper. Why?

2. The assumption of label-rich advantage looks somewhat artificial. What was the motivation of singling out this assumption? Is this assumption backed up with empirical evidence?

**Ethical Concerns:**

["NO or VERY MINOR ethics concerns only"]

**Final Justification:**

The authors have addressed my technical concerns, and proposed a plan to improve the writing in the camera ready version. Therefore, I raised my rating from 3 to 4.

**Limitations:**

Yes.

**Paper Formatting Concerns:**

No.

**Quality:**

3

**Strengths And Weaknesses:**

**Strength**

This paper is overall well-written. It provides a simple yet effective solution to detect malicious client after deployment. The experiments cover a decent number of research questions and the designs are fair.

**Weakness**

The theoretical analysis in Section 5 is unnecessary. The proof are very definitional from the assumptions. I suggest removing it and shifting more important results such as how to determine if a misclassified input is target or non-target.

Overall, I find the idea of this paper solid and the approach sound. The setting may not be the most interesting, but it seems to be the first paper investigating this problem. The empirical evaluation is the highlight while some of the theoretical claims and assumptions seem unnecessary to me. The paper is borderline. I want to listen to authors' response to my questions.

---

> ### Author Rebuttal · Authors · 2025-07-30
>
> Thank you for your feedback! We will carefully consider the theoretical analysis section and adjust the content of the paper accordingly.
>
> Q1: **Results is different from the original paper**: Our experimental setup is largely aligned with that of FLTrust, with two main differences: the amount of data held by the server and the scaling factor used in the Scaling attack. In the FLTrust paper, the server holds 100 data samples, and the scaling factor is set to the number of clients (100 in their experiments). This means each malicious update is scaled by a factor of 100 to amplify its influence during aggregation. In contrast, our work uses only 50 server data samples and sets the scaling factor to 1. This design choice is motivated by the following considerations:
>
> 1). We found that when the server holds 100 data samples, the ASR of the Scaling attack drops to 0.01, which aligns with the results reported in the FLTrust paper. In such cases, using FLForensics to trace back malicious clients and retrain a global model becomes less meaningful. Therefore, we reduce the number of server data samples to make the attack more effective.
>
> 2). Setting the scaling factor to 1 avoids making the malicious updates significantly larger in magnitude than benign updates. If we used a scaling factor of 100, the malicious updates would become trivially detectable, for instance, by simply filtering out updates with abnormally large norms. To demonstrate that FLForensics can accurately trace back malicious clients even when malicious updates appear similar in scale to benign ones, we chose a scaling factor of 1.
>
> Additionally, we also conducted experiments using the same settings as FLTrust. The results are as follows:
>
> 100 server data samples and a scaling factor of 1:
>
> |Attack|TACC|ASR|DACC/FPR/FNR|
> |-|-|-|-|
> |Scaling|0.815|0.012|1.000/0.000/0.000|
> |ALIE|0.827|0.028|1.000/0.000/0.000|
> |Edge|0.823|0.036|0.920/0.000/0.000|
>
> 100 server data samples and a scaling factor of 100:
>
> |Attack|TACC|ASR|DACC/FPR/FNR|
> |-|-|-|-|
> |Scaling|0.815|0.031|1.000/0.000/0.000|
> |ALIE|0.826|0.037|1.000/0.000/0.000|
> |Edge|0.803|0.046|0.980/0.000/0.100|
>
> The results show that FLForensics can still accurately trace back malicious clients. Although the detection accuracy for certain attacks (e.g., Edge attack) is lower in these settings, this is largely because such attacks have negligible impact. This highlights a trade-off between attack effectiveness and traceback effectiveness.
>
> Q2: **Label-rich advantages**: Our motivation for the "label-rich advantage" assumption stems from the empirical observation that a subset of benign clients consistently exhibits very high influence scores. Figure 2(a) illustrates this phenomenon: clients are color-coded as red (malicious), green (Category I benign), and blue (Category II benign). As shown, the influence scores of Category I benign clients are very close to those of malicious clients, while Category II benign clients display significantly different influence patterns. This also explains why **FLForensics-A**, which uses only the one-dimensional influence score, consistently exhibits some FPR. Since the 1D influence score cannot distinguish between malicious clients and Category I benign clients, it tends to misclassify Category I benign clients as malicious.
>
> To further validate this assumption, we conducted the following experiment: we relabeled all test data samples to the target label and evaluated the cross-entropy loss on these inputs using each client’s final-round local model. Averaging the results, we found that malicious clients had an average loss of 4.82, while Category I and Category II benign clients had average losses of 0.18 and 12.04, respectively. This indicates that **Category I clients are far more likely to predict arbitrary inputs as the target label**, even more so than the malicious clients themselves. This clear separation in behavior provides strong empirical support for the "label-rich advantage" assumption.

---

> > ### Comment · Reviewer_Y9xs · 2025-08-05
> > **Post Rebuttal**
> >
> > I appreciate the experiment and clarification from the authors.
> >
> > The intuition of the approach is sound. I have no particular concern of the success of this approach. However, the presentation of the paper can be somewhat misleading. For example, I understand the assumption of Label-rich advantage and its root in the empirical observation. However, is it necessary to create a jargon like label-rich advantage? The advantage of whom over what? This name is not helping but hurting reader's understanding. Similarly, the "theorems" in the main body are rather definitional given the assumptions. I would prefer down-tone it to observations. I personally prefer a simple idea that works, which is exactly what this paper achieves, than adding unnecessary terminology and theorems that distract the reader from reading.
> >
> > This paper is borderline for me and it has clear strength (a sound approach) and weakness (its presentation style) to me. I will engage the discussion with fellow reviewers and AC to see if the strength outweighs the weakness.

---

> > > ### Author Response · Authors · 2025-08-05
> > >
> > > Thank you for your recognition of our approach. We fully agree with your point that a simple yet effective method improves readability and understanding for readers. This is precisely why we designed a relatively lightweight algorithm and intentionally limited the length of the theoretical analysis in the main paper (just over half a page).
> > >
> > > We included the theorem section to provide a theoretical perspective on why a two-dimensional influence score is necessary to distinguish different types of benign clients, rather than relying solely on empirical evidence. The theoretical justification relies on certain assumptions, which are either supported by our empirical findings (e.g., Label-rich Advantage) or commonly adopted in the literature (e.g., Local Linearity).
> > >
> > > Regarding the Label-rich Advantage assumption, our intuition was from the observation that a client with a larger number of samples from a specific label is naturally more likely to predict that label correctly, which constitutes an advantage over clients that lack such exposure. This observation motivated the assumption that clients possessing more data with the target label are more likely to predict the target label. That said, we acknowledge that the term “advantage” may be heuristic and potentially misleading to some readers.
> > >
> > > We sincerely appreciate your suggestion and are happy to revise the name of the assumption accordingly. We are also willing to move the theorem part to the appendix and reduce its prominence in the main text, as recommended.

---

> ### Author Response · Authors · 2025-08-04
>
> Thank you for your response. We would like to ask whether our responses have fully addressed your concerns. If there are any remaining questions or suggestions, we would be happy to provide further clarification.

---

### Official Review · Reviewer_Fzy5 · 2025-07-03

**Clarity:** 3
**Significance:** 3
**Originality:** 4
**Rating:** 4
**Confidence:** 4

**Summary:**

This paper proposes FLForensics, the first post-hoc forensic method for identifying malicious clients in FL after a poisoning attack has occurred. Instead of preventing attacks during training, FLForensics traces back attackers once a misclassified target input is discovered from the deployed global model. It assigns each client a 2D influence score based on how their updates affect the model’s behavior on both the target input and a non-target input. The paper also offers both theoretical and empirical justifications.

**Questions:**

1. While the method is in general interesting, some of the details seem bizarre, for example, the method requires "scaled" distance to work seemingly suggest the method itself is not robust enough. Can the authors demonstrate why the scaling is necessary? For example, are there cases where non-scaled cases can perform better for clustering?

2. Intuitive speaking, the results should highly depend on the strength of attacks, probably whether these malicious clients are all targeting the same target. How did the authors choose the strength of the attacks?
    - for the attacks that are strong enough to flip 100% of the labels, the detection might be able to identify, but what if the clients are minor and only want to flip the label in a lower rate?
   - are there trade-offs between the success rate of the attack vs. the success rate of the detections?
   - line 237 says the paper will also present ASR, but the table does not seem to have these results

3. while the local linearity assumption is fairly reasonable in ML community, in this particular setting, the authors are referring to the \delta as model updates, thus the problem might depend on the practical settings on the FL system and how the system chooses to update the model, some discussions, and evidence that in most FL settings, this assumption still holds might be necessary.

**Ethical Concerns:**

["NO or VERY MINOR ethics concerns only"]

**Final Justification:**

The reviewer has mostly addressed my concerns, I will keep my original positive score.

**Limitations:**

There is no explicitly section on limitation, but the authors have discussed future work, which can probably be interpreted as limitations.

**Paper Formatting Concerns:**

None.

**Quality:**

3

**Strengths And Weaknesses:**

strengths

- The novel problem setting is very interesting, practically important, and a creative usage of the influence function method.

- The emperiments have been conducted over a range of datasets and model architectures, although fairly small, but reasonable in FL studies.

weakness:

- The method requires access of target input and also access to the client model after training to "tests every client on a random non-target input 50 with the target label". In practice (not experimental settings), when the server relies client on a practical manner, the requirement does not seem to be very realistic.

- The method also relies on unsupervised manner (clustering) to detect anormally, through which the results seem too ideal (see questions for more details)

---

> ### Author Rebuttal · Authors · 2025-07-30
>
> Thank you for your recognition of our work and your valuable comments. Below are our responses.
>
> W1: **Requirements seems not realistic**: FLForensics traces back malicious clients based on a single reported misclassification event. We assume the server has received such an input from a client. To access client models, we store model checkpoints from selected rounds during training, eliminating the need to request any additional information post-training. In summary, the only requirement is a misclassification event reported by a client.
>
> Furthermore, as discussed in Appendix G (“Detecting Misclassified Target Input”), our method can distinguish between misclassifications caused by poisoning attacks and those resulting from natural errors. Therefore, the misclassified input does not need to be a poisoning-induced target input.
>
> W2: **Relies on unsupervised manner**: Please see the responses for Q1, Q2, and Q3.
>
> Q1: **Why scaling is necessary**: The scaling operation enhances the robustness of our method. As illustrated in Figure 2(b), since $s_i$​ and $s’_i$ may be on different scales, clustering can become significantly biased, placing excessive weight on the dimension with larger magnitudes (typically $s’_i$​) while largely ignoring the other. As a result, clusters (i.e., points with different shapes) are formed primarily based on $s’_i$, leading to inaccurate separation. By scaling the two influence scores, clustering considers both dimensions equally, regardless of their original scales. In other words, the scaling operation ensures that FLForensics can robustly identify the malicious client cluster even when $s_i$ and $s’_i$ are on different scales. Therefore, the scaled case is consistently more stable and robust than the non-scaled case.
>
> Q2: **The strength of the attack**: (1). As shown in Figure 4, the Adaptive Attack achieves very high ASR under some settings (e.g., attack frequency = 1, attack probability = 1, trigger size = 10, and trigger value is random or white), while the ASR is relatively low under others (e.g., attack frequency = 10, attack probability = 0.2, etc.). When the attack is highly effective, we achieve nearly 100% detection accuracy. Even when the attack is less effective, our method still maintains over 90% detection accuracy in most cases. The only exceptions arise when the trigger value is black or the trigger size is 2$\times$2, in which cases the detection accuracy drops. However, in these scenarios, the ASR is 0, indicating that the attacks are ineffective.
>
> (2). According to Figure 4, there is generally a trade-off: higher ASR typically makes the attack easier to detect. However, even in many scenarios with very low ASR (e.g., attack frequency = 10, trigger location = UL, trigger value = grey, trigger size = 3$\times$3), our method still achieves nearly 100% detection accuracy.
>
> (3). ASR is an important metric. However, due to space limitations, we had to place the ASR results in Table 6 of the Appendix. We mention this on line 263. We will add a citation to Table 6 at line 237.
>
> Q3: **Problem might depend on the practical settings**: In common FL systems, clients exchange either models or model updates with the server. In our paper, we assume that model updates are exchanged. However, if full models are shared instead, the server can still compute the updates by subtracting the previous global model (e.g., $w^{(i)}_ {t+1} - w_ t$, where $w^{(i)}_ {t+1}$ is the shared model from client $i$.) allowing influence scores to be calculated for traceback. Additionally, the server can apply different aggregation rules when updating the global model. We demonstrate this in Figure 3(c), which shows results under various aggregation rules. These observations collectively indicate that FLForensics remains effective under different FL system settings.

---

> > ### Comment · Reviewer_Fzy5 · 2025-08-05
> > **re rebuttal**
> >
> > Thanks for the rebuttal. I will keep my score as it's already positive.

---

> > > ### Author Response · Authors · 2025-08-05
> > >
> > > Thank you for your response. We truly appreciate your recognition of our work and your thoughtful review.

---

### Decision · Program_Chairs · 2025-09-17

**Decision:**

Accept (poster)

**Comment:**

This paper received the following ratings: Borderline accept, Borderline accept, Reject, Accept. The authors introduce FLForensics, a post-hoc forensic method for identifying malicious clients in federated learning after a poisoning attack. By computing 2D influence scores from stored model updates, it traces back attackers once a misclassified input is detected. Most reviewers find the proposed method novel and practically important. The experiments have been conducted over a range of datasets and model architectures, showing strong detection accuracy. The proposed approach is also accompanied by a formal analysis. Some concerns are also raised about some unrealistic assumptions and lack of comparison to more advanced defenses. AC finds the strength outweighs the weakness, and recommends accepting the paper.